# Cell-Based Influenza A/H1N1pdm09 Vaccine Viruses Containing Chimeric Hemagglutinin with Improved Membrane Fusion Ability

**DOI:** 10.3390/vaccines8030458

**Published:** 2020-08-19

**Authors:** Madoka Kawahara, Toshiya Wada, Fumitaka Momose, Eri Nobusawa, Yuko Morikawa

**Affiliations:** 1Kitasato Institute for Life Sciences, Kitasato University, Tokyo 108-8641, Japan; kawahara@lisci.kitasato-u.ac.jp (M.K.); to_shi_ya_04_16@yahoo.co.jp (T.W.); fmomose@lisci.kitasato-u.ac.jp (F.M.); 2Department of Virology I, National Institute of Infectious Diseases, Tokyo 162-8640, Japan; 3Influenza Virus Research Center, National Institute of Infectious Diseases, Tokyo 208-0011, Japan; nobusawa@nih.go.jp

**Keywords:** influenza, H1N1 2009, cell-based vaccine, reverse genetics, chimeric, PR8

## Abstract

The H1N1 influenza pandemic vaccine has been developed from the A/California/07/09 (Cal) virus and the well-known high-yield A/Puerto Rico/8/34 (PR8) virus by classical reassortment and reverse genetics (RG) in eggs. Previous studies have suggested that Cal-derived chimeric hemagglutinin (HA) and neuraminidase (NA) improve virus yields. However, the cell-based vaccine of the H1N1 pandemic virus has been less investigated. RG viruses that contained Cal-derived chimeric HA and NA could be rescued in Madin–Darby canine kidney cells that expressed α2,6-sialyltransferase (MDCK-SIAT1). The viral growth kinetics and chimeric HA and NA properties were analyzed. We attempted to generate various RG viruses that contained Cal-derived chimeric HA and NA, but half of them could not be rescued in MDCK-SIAT1 cells. When both the 3′- and 5′-terminal regions of Cal HA viral RNA were replaced with the corresponding regions of PR8 HA, the RG viruses were rescued. Our results were largely consistent with those of previous studies, in which the N- and C-terminal chimeric HA slightly improved virus yield. Importantly, the chimeric HA, compared to Cal HA, showed cell fusion ability at a broader pH range, likely due to amino acid substitutions in the transmembrane region of HA. The rescued RG virus with high virus yield harbored the chimeric HA capable of cell fusion at a broader range of pH.

## 1. Introduction

The new H1N1 influenza virus emerged in 2009 and has rapidly spread worldwide. This virus, termed A/California/07/09 (H1N1) (designated as Cal), has a unique feature in terms of genetic combination: five viral RNA (vRNA) segments of swine origin (for hemagglutinin (HA), nucleoprotein (NP), neuraminidase (NA), matrix (M), and non-structure), two of avian origin (for PB2 and PA), and one of human origin (for PB1). Although the pathogenicity and virulence of this virus were relatively mild, the HA was genetically and antigenically different from that of the previous H1N1 seasonal virus, suggesting the need for a vaccine for the new H1N1 virus. Unfortunately, this virus grew very poorly in eggs, which were generally used for the vaccine manufacturing process.

Substantial effort has been made to generate vaccine seed viruses using classical reassortment or reverse genetics (RG) technology. The A/Puerto Rico/8/34 (H1N1) virus (designated as PR8) has been used as a backbone virus because it is a well-known high-yield strain in eggs and laboratory cell lines, and it is highly attenuated in humans. The NYMC-X179A vaccine virus was generated by coinfection with the Cal and PR8 viruses into eggs, followed by selection with anti-PR8 HA and NA antisera [1]. The NIBRG-121 vaccine virus has been generated using RG, and it contains the HA and NA vRNAs from Cal and the other vRNAs from PR8. This virus was further passaged in eggs and Vero cells to produce NIBRG-121xp (extended passage), which was similar to the reassortant viruses X-181 and X-181A in showing a higher HA titer, indicating improved HA yield in the vaccine seed viruses [1]. However, nucleotide (nt) sequencing revealed several amino acid mutations in HA in these high-yield viruses (X-181, X-181A, and NIBRG-121xp), suggesting that they were egg-adaptive mutations generated by serial passage in eggs [1].

Further studies focused on chimerization of HA to improve viral growth and/or HA yield. The RG viruses that harbored chimeric HA molecules with the PR8 transmembrane (TM) and cytoplasmic tail (CT) domains, NIBRG-118 and NIBRG-119, displayed higher HA yields in eggs than the non-chimeric NIBRG-121 [2]. This was also the case for H5N1 RG viruses that comprised HA and NA derived from A/Vietnam/1194/2004 (H5N1) on a PR8 backbone [3]. The effects of chimeric NA molecules were also explored in virus reassortment systems. The mutant reassortant virus that harbored chimeric NA with the CT-TM-stalk (ST) regions of PR8 and the globular head region of A/Texas/05/09 (H1N1), compared with the original reassortant virus, showed a slightly increased HA yield with similar growth kinetics in Madin–Darby canine kidney (MDCK) cells [4]. A similar chimeric NA was also constructed in the reassortant virus from the A/Vietnam/1194/2004 (H5N1) and PR8 viruses, resulting in a higher virus yield in eggs and MDCK cells than that observed for the non-chimeric reassortant virus [5]. Finally, the bi-chimeric virus that harbored chimeric HA and NA was generated from the NYMC-X179A virus, and it showed a higher HA yield than NYMC-X179A [6]. These studies suggested that both HA and NA chimeras provided the benefit of higher HA yield and virus titers. However, the mechanistic rationale for this benefit remains to be elucidated.

HA forms a homotrimer, while NA forms a homotetramer. HA trimerization is important for intercellular membrane fusion in viral infection [7,8]. However, some studies have suggested that the trimer of Cal HA, compared to those of the HA of other H1N1 seasonal viruses, is relatively unstable [9,10]. Studies have also shown that Cal HA is more sensitive to pH and temperature than the HAs of other H1 strains, and thus, it is easily digested by trypsin [9,11]. Independent studies have explored HA trimerization and suggested that amino acid mutations in the HA2 subunit stabilized HA trimer formation and improved viral growth ability [10,12,13]. Studies have also shown that the mutations in HA2 confer higher thermal stability membrane fusion ability at a wider range of pH to HA. From these studies, it can be concluded that Cal HA is most likely structurally unstable and produces a lower virus yield than the HAs of other H1 viruses. For vaccine development, the RG virus, in which the TM-CT region of HA was derived from PR8 and the other HA regions were from Cal, was generated, showing improvement in virus yield [2]. However, the properties of the chimeric HA have not yet been fully investigated.

Reassortant viruses generated by a mixed infection of two viruses, e.g., Cal plus PR8, often appear to have HA, NA, and PB1 vRNA segments from one virus and other segments from the other virus, suggesting that some vRNA segments may associate with each other during vRNA reassortment [14]. Each vRNA segment forms a panhandle structure via base pairing of its 3′- and 5′-end sequences, and a supramolecular complex of eight distinct vRNA segments is further formed, most likely through segment–segment interactions [15,16,17]. RG studies with vRNA segments, in which large portions of the coding regions were replaced by reporter genes, have revealed segment-specific packaging signals present in both the 3′- and 5′-terminal regions of each vRNA segment (50–250 nt), in not only the noncoding regions (NCRs), but also the terminal coding regions [18,19,20]. This makes it more complicated to successfully generate RG viruses, especially when they contain chimeric vRNA segments.

For the H1N1pdm09 vaccine candidate, several viruses that harbored Cal-derived chimeric HA and NA have been generated in eggs; however, cell-based H1N1pdm09 vaccines have been less investigated, despite their ability to avoid egg adaptations and supply dependency. Here, we used cell culture systems and attempted to generate various RG viruses that harbored Cal-derived chimeric HA and NA segments on a PR8 backbone. Furthermore, we analyzed the rescued RG viruses and the properties of their HA/NA molecule. Our data confirmed the importance of the 3′ and 5′ NCRs of HA vRNA but revealed that amino acids in the TM of HA contributed to the trimerization and cell fusion ability of HA. Our systemic analysis provided insights into the production of cell-based H1N1 vaccines using RG technology.

## 2. Materials and Methods

### 2.1. DNA Construction and RG

The coding sequence (CDS) of human β-galactoside α2,6-sialyltransferase (SIAT1) was linked in frame with the CDS of downstream hygromycin B phosphotransferase (HygR) through the *Thosea asigna* virus 2A-like self-processing sequence and then, cloned into the mammalian expression plasmid pCAGGS [21]. Briefly, SIAT1 CDS was amplified from reverse transcripts of the 293T cell RNA by PCR with a single forward primer (IFGGSXhoKz-SIAT1-F, 5′-CAAAGAATTCCTCGAGCCACCATGATTCACACCAACCTGAAGAAAAAG-3′) and a mixture of two reverse primers (SIAT1-2A1GRGSct-R, 5′-AGGCTGCCTCTGCCCTCAGATCCGCAGTGAATGGTCCGGAAGCCAG-3′, and 2A1-gGRGS-rev, 5′-AGGGCCAGGGTTTTCCTCCACGTCGCCGCAGGTCAGCAGGCTGCCTCTGCCC-3′). The HygR CDS, followed by the FLAG epitope tag CDS, was amplified from the pEBMulti-Hyg vector (Wako Pure Chemicals, Osaka, Japan) by PCR with a single forward primer (IF2A1ENPGP-HygR-F, 5′-GAAAACCCTGGCCCTATGAAAAAGCCTGAACTCACCG-3′) and a mixture of two reverse primers (HygR-FlgDDDK-R, 5′-CTTGTCGTCGTCATCCTTGTAGTCGTTAGCCTCCCCCATCTCC-3′, and IFGGS-FlgKDDDDKstpNhe-R, 5′-GACCACCCAAGCTAGCTACTTGTCGTCGTCATCCTT-3′). These two DNA fragments were assembled using the In-Fusion Cloning System (Clontech Laboratories, Takara, Kusatsu, Japan) to generate a polycistronic expression plasmid pCAGGS-SIAT1-HygR-F.

The HA and NA CDSs of influenza viruses PR8 and Cal were amplified by PCR and cloned into pCAGGS. The cDNAs for chimeric HA, composed of an external HA domain from Cal HA and the TM and cytoplasmic domains from PR8 HA, were constructed by overlapping PCR and cloned into pCAGGS. The cDNA for chimeric NA, composed of the cytoplasmic, TM, and ST domains from PR8 NA and the globular head domain from Cal NA, was similarly constructed.

For the recovery of RG viruses with chimeric HA and NA, chimeric HA and NA cDNAs, which contained the terminal NCRs of PR8 or Cal, were similarly constructed by PCR. They were cloned into the mammalian RNA expression plasmid pHH21, which contained the human RNA *polymerase I* promoter and mouse RNA *polymerase I* terminator sequences.

The RG system for the influenza virus has been described previously [22]. Briefly, 293T cells were transfected with eight pHH21 plasmids for the synthesis of each vRNA segment and four pCAGGS plasmids for the expression of PR8 PB1, PB2, PA, and NP. The culture medium was replaced with Opti-minimum essential medium (MEM) I (Gibco, Thermo Fisher Scientific, Waltham, MA, USA) supplemented with acetyl trypsin (5 μg/mL) and 0.3% bovine serum albumin (BSA) 6 h post transfection, and the cells were incubated for 2 days. Recovered viruses were grown in MDCK cells overexpressing SIAT1 (MDCK-SIAT1, see below). MDCK-SIAT1 cells were infected with RG viruses for 1 h and were incubated in Opti-MEM I supplemented with acetyl trypsin (5 μg/mL) and 0.3% BSA at 34 °C for 3–5 days.

### 2.2. Cell Culture, Virus Infection, and DNA Transfection

The preparation of MDCK-SIAT1 cells has been described previously [23]. We independently established MDCK-SIAT1 cells via transfection of a SIAT1 expression plasmid pCAGGS-SIAT1-HygR-F and subsequent selection with hygromycin (400 μg/mL) for a week. The expression of exogenous SIAT1 in nearly all cells was confirmed by immunofluorescence staining with anti-TaV2Apep antibody (Ab). The established MDCK-SIAT1 cells were maintained in the absence of hygromycin B and then, used for virus infection. MDCK, MDCK-SIAT1, and 293T cells were grown in Dulbecco’s modified Eagle’s medium (DMEM; Sigma-Aldrich, St. Louis, MO, USA) supplemented with 10% fetal bovine serum (FBS) at 37 °C under 5% CO_2_. MDCK and MDCK-SIAT1 cells were infected with RG viruses in Opti-MEM I supplemented with 0.3% BSA for 1 h and incubated in Opti-MEM I supplemented with acetyl trypsin (5 μg/mL) and 0.3% BSA at 34 °C. For protein analysis, 293T cells were transfected with HA and NA expression plasmids and incubated at 37 °C for 24 h. In some experiments, 293T cells were cotransfected with HA/NA and EGFP expression plasmids.

### 2.3. Virus Growth Kinetics and Titration

MDCK-SIAT1 cells were seeded into 12-well plates (2 × 10^5^ cells/well) and infected with RG viruses in Opti-MEM I supplemented with 0.3% BSA for 1 h. After washing, the cells were incubated in Opti-MEM I supplemented with acetyl trypsin (5 μg/mL) and 0.3% BSA at 34 °C for 5 days. The culture medium was harvested every 12 h and subjected to virus titration assays using MDCK-SIAT1 cells.

For the plaque assay, the culture medium was 10-fold serially diluted with Opti-MEM I supplemented with 0.3% BSA and inoculated into MDCK-SIAT1 cells at 37 °C for 1 h. The cells were overlaid with 0.6% agarose in MEM Eagle’s medium (without phenol red, Thermo Fisher) supplemented with acetyl trypsin (5 μg/mL) and incubated at 34 °C for 2–3 days. The cells were fixed with a 1:1 mixture of acetic acid and ethanol for 2 h. After protein blocking, the cells were incubated with rabbit anti-NP Ab for 1 h. The cells were washed with phosphate-buffered saline (PBS) containing 0.1% Tween20 and incubated with peroxidase-conjugated anti-rabbit IgG Ab. Plaques were visualized with the TrueBlue Substrate (KPL, Seracare Life Sciences, Milford, MA, USA).

### 2.4. Virus Concentration

MDCK-SIAT1 cells were infected with RG viruses at a multiplicity of infection (MOI) of 1.0 and incubated at 34 °C for 1 or 2 days. The culture medium (10 mL) was clarified at 7200× *g* at 4 °C for 30 min. Viruses were pelleted through 20% sucrose cushions by centrifugation at 175,000× *g* at 4 °C for 90 min. Virus pellets were resuspended in PBS (100 μL).

### 2.5. SDS-PAGE, Western Blotting, and Blue Native (BN)-PAGE

Protein samples were subjected to SDS-PAGE and then, transferred onto a PVDF membrane. The membrane was incubated with rabbit polyclonal anti-H1 HA, mouse monoclonal anti-Cal HA external domain (11055-MM08, Sino Biological, Beijing, China), rabbit monoclonal anti-PR8 HA external domain (aa 1–528, 11684-R07, Sino Biological), sheep anti-NA external domain (aa 37–469, AF4858, R&D Systems, Minneapolis, MN, USA), and rabbit anti-NP Abs. In some experiments, the membrane was also probed with mouse anti-Actin (Sigma-Aldrich), anti-caveolin (Abcam, Cambridge, UK), anti-GFP (Clontech, BD Biosciences, Franklin Lakes, NJ, USA), and anti-FLAG (M2, Sigma-Aldrich) monoclonal Abs (mAbs). After washing with PBS containing 0.1% Tween20, the membrane was incubated with peroxidase-conjugated anti-mouse, anti-sheep, or anti-rabbit IgG Ab. The membrane was developed using ECL prime reagents (GE Healthcare, Chicago, IL, USA). In some experiments, the band intensities in the blots were semi-quantified using ImageJ software. For purified particles, viral particle levels were normalized to NP levels in viral particle fractions, and HA and NA levels relative to NP levels were calculated.

For protein oligomerization, protein samples were incubated in 25 mM MES buffers (pH, 4.5–6.5) for 30 min. After brief sonication, the lysates were treated with 1% Triton X-100 at 37 °C for 30 min and centrifuged at 17,400× *g* for 5 min. BN-PAGE was performed using 4–12% gradient polyacrylamide gels [24].

### 2.6. Triton X-100 Solubilization Analysis

Briefly, 293T cells were seeded into 12-well plates (5 × 10^5^ cells/well) and transfected with HA and NA expression plasmids. The cells were then resuspended with cold TNE buffer (50 mM Tris-HCl (pH 7.5), 1 mM EDTA, and 150 mM NaCl) containing 1 mM dithiothreitol and protease inhibitor cocktail (Roche, Basel, Switzerland) 24 h post transfection. After brief sonication, the lysates were treated with 1% Triton X-100 at 4 °C for 30 min and centrifuged at 17,400× *g* at 4 °C for 30 min. The soluble (supernatant) and insoluble (precipitate) membrane fractions were subjected to SDS-PAGE, followed by Western blotting.

### 2.7. Cell Fusion Assay

The cell fusion assay has been described previously [12,13]. Briefly, 293T cells were cotransfected with HA expression plasmids and an EGFP expression plasmid and then, incubated at 37 °C for 24 h. After washing with PBS twice, the cells were treated with acetyl trypsin (5 μg/mL) at 37 °C for 5 min. Trypsin was inactivated by washing with PBS containing 1% FBS. For cell fusion, the cells were treated with acidic PBS (pH, 5.0–7.0, adjusted with citric acid) for 1 min. After washing with PBS, the cells were incubated in growth medium supplemented with 10% FBS at 37 °C for 2 h. The cells were fixed with 2% paraformaldehyde and cell nuclei were stained with DAPI. The cells were observed by confocal microscopy.

### 2.8. NA Activity Assay

The NA activity was measured by a fluorometric assay with substrate 2′-(4-methylumbelliferyl)-α-d-*N*-acetylneuraminic acid (MUNANA) (ab138888, Abcam). Samples (cells expressing NA constructs and RG viruses) were lysed with 0.1% Triton X-100. After centrifugation at 17,500× *g* at 4 °C for 10 min, the supernatants were subjected to an NA assay at 37 °C for 1.5 h. Reaction products were measured at 365 nm for excitation and 450 nm for emission. NA activity was presented as the NA enzymatic activity (unit) that produces 4-MU (μmole) of the product.

### 2.9. Confocal Microscopy

MDCK-SIAT1 cells were infected with RG viruses at a MOI of 1.0 and incubated at 34 °C for 1 or 2 days. The cells were fixed with 2% paraformaldehyde, and their membranes were permeabilized with 0.5% Triton X-100 in PBS. The cells were incubated with rabbit anti-H1 HA and sheep anti-NA Abs and subsequently, Alexa Fluor 488- and Alexa Fluor 568-conjugated secondary Abs (Molecular Probes, Thermo Fisher Scientific). Cell nuclei were stained with DAPI. The cells were observed with a confocal microscope (TCS-SP5II, Leica Microsystems, Wetzlar, Germany), and 293T cells were similarly analyzed by confocal microscopy for cell fusion assays.

## 3. Results

### 3.1. Construction and Expression of Chimeric HA and NA

Previous studies have indicated that RG viruses with chimeric HA vRNA, NIBRG-118, and NIBRG-119, in which the TM-CT regions are replaced by the corresponding regions of PR8 HA, show higher viral growth in eggs than NIBRG-121, which comprises the Cal HA and NA vRNAs on the PR8 backbone [2]. The chimeric NA-harboring viruses, in which the NCR and CT-TM-ST regions are derived from PR8 and the globular head region from the 2009 pandemic H1N1 virus, showed an increased HA yield [4,6]. Based on these data, we similarly constructed chimeric HA and NA (Figure 1A). One chimeric HA, designated as Cal-P HA (corresponding to the HA of NIBRG-118), possessed an external HA domain, including the signal peptide, from Cal HA (aa 1–529) and the TM-CT domains from PR8 HA (aa 530–566). Another chimeric HA, designated as P-Cal-P HA (corresponding to the HA of NIBRG-119), possessed an external HA domain from Cal HA (aa 19–529) and the signal peptide and TM-CT domains from PR8 HA (aa 1–18 and aa 530–566). A chimeric NA, designated as P-Cal NA (corresponding to the NAst in a previous study [4]), possessed the CT-TM-ST domains from PR8 NA (aa 1–75) and the globular head of NA from Cal HA (aa 76–454). The ST region of PR8 NA was 15 amino acids shorter than that of Cal NA.

These constructs were cloned into the expression plasmid pCAGGS and expressed in 293T cells. Protein expression was analyzed by SDS-PAGE, followed by Western blotting (Figure 1B). Western blotting with anti-actin and anti-GFP mAbs and subsequent quantification of the band intensities with ImageJ software confirmed that the loading sample levels and the transfection efficiencies were nearly equivalent. When the blots were probed with rabbit anti-H1 HA polyclonal Ab and the HA band intensities, the HA expression levels were also found to be comparable. The use of mouse anti-Cal HA and rabbit anti-PR8 HA mAbs, both of which specifically recognized their HA external domains, confirmed that Cal-P HA and P-Cal-P HA were reactive with anti-Cal HA mAb, but not anti-PR8 HA mAb. When NA expression was similarly analyzed with anti-NA polyclonal Ab, the expression levels of Cal NA and P-Cal NA were significantly lower than that of PR8 NA; however, P-Cal NA expression, compared with Cal NA expression, was slightly improved.

### 3.2. Rescue of RG Viruses Containing Chimeric HA/NA

The chimeric HA and NA segments were cloned into the RNA expression plasmid pHH21. The eight pHH21 plasmids for the synthesis of each vRNA and the four pCAGGS plasmids for the expression of PR8 PB1, PB2, PA, and NP were cotransfected into 293T cells. The RG viruses were amplified in MDCK or MDCK-SIAT1 cells. In general, RG viruses could be more easily rescued in MDCK-SIAT1 cells, and this was in agreement with the findings of previous studies [23,25]. We did not use eggs to avoid unwanted mutations.

Combinations of chimeric HA and NA tested in this study are shown in Figure 2. The nt alignments of both the 3′- and 5′-terminal regions of the PR8 and Cal strains are shown in Figure 3. The RG virus that harbored Cal HA and Cal NA on the PR8 backbone could not be rescued in our 293T/MDCK-SIAT1 cell systems. This failure was possibly due to the absence of Cal PB1, as observed in the 5:3 reassortant virus [1]; however, NIBRG-121, the RG virus with 6:2 genetic constellation, was rescued in eggs [2]. Similarly, the RG virus that harbored Cal HA alone on the PR8 backbone could not be rescued. In contrast, the RG virus that harbored Cal NA was rescued, suggesting that an HA segment was more responsible for the rescue of the RG virus. This is likely due to the higher nt diversity in HA NCR than in Cal NCR when the PR8 and Cal strains were compared (Figure 3). We attempted to rescue the RG viruses that harbored chimeric HAs on the PR8 backbone (Figure 2). The “Cal-P/PR8” virus harbors one-side chimeric HA, in which the 3′-NCR-signal peptide region and the external domain are derived from Cal and the TM-CT-5′-NCR is from PR8 and PR8 NA. The “Cal-P/Cal” virus harbors the chimeric HA and Cal NA. Thus, the HA and NA constructs in the RG virus are shown with a slash hereafter. These two RG viruses could not be rescued. However, when the chimeric HA was replaced by both-side chimeric HA, in which the 3′-NCR-signal peptide region and the TM-CT-5′-NCR were derived from PR8 and the external domain were from Cal, the RG viruses could be rescued, regardless of whether NA was derived from PR8 or Cal (see P-Cal-P/PR8 and P-Cal-P/Cal viruses). These results suggested that the constructs, in which both the 3′ and 5′ NCRs of the HA vRNA segment and their terminal coding regions were derived from PR8, were preferable for the rescue of RG viruses with a PR8 backbone.

We similarly explored combinations of a chimeric NA, in which the 3′-NCR-CT-TM-ST region was derived from PR8 and the head domain-5′-NCR was from Cal (designated as P-Cal for NA) with various HAs (Figure 2). When this chimeric NA was combined with PR8 HA, the RG virus PR8/P-Cal was rescued. In contrast, when the chimeric NA was combined with Cal HA, the RG virus Cal/P-Cal was not rescued. As expected, when the chimeric NA was combined with the one-side chimeric HA, the RG virus Cal-P/P-Cal could not be rescued. However, when the HA segment was replaced by the both-side chimeric HA, the RG virus P-Cal-P/P-Cal was rescued, suggesting that the presence of both 3′ and 5′ NCRs of PR8 HA vRNA and their terminal coding regions was preferable, even in combination with the chimeric NA.

Since the segment-specific packaging signals extend to both the 3′- and 5′-terminal coding regions of the vRNA segments [18,19,20], we also added approximately 100~115 nt of the 3′- and 5′-terminal regions of the PR8 HA/NA segment to the HA/NA coding region to ensure segment packaging (Figure 2). The 3′-terminal vRNA region (105 nt) added to the HA coding region was longer than the 3′-NCR-signal peptide region, thereby ensuring segment packaging. The start codon (methionine) in the additional region was replaced by a stop codon to suppress translation. When the 3′-terminal (105 nt) and 5′-terminal (100 nt) regions of the PR8 HA segment were added to the coding region of the both-side chimeric HA (designated as 105(P-Cal-P)100 for HA) and combined with the PR8 NA segment, the RG virus 105(P-Cal-P)100/PR8 could not be rescued. This result suggested that not only the terminal regions but also the total nt length of the HA segment was possibly critical for segment packaging because the P-Cal-P/PR8 virus, which did not harbor the additional terminal regions in the HA segment, could be rescued. In contrast, when the 3′-terminal (115 nt) and 5′-terminal (100 nt) regions of the PR8 NA segment were added to the coding region of Cal NA (designated as 115(Cal)100 for NA), the total nt length of NA segment was tolerant to segment packaging (comparing the PR8/Cal and PR8/115(Cal)100 viruses). The 115(Cal)100 NA segment did not rescue the Cal HA segment but did rescue the Cal-P HA segment. Considering that the authentic Cal NA segment did not rescue the Cal HA or Cal-P HA segment, it could be suggested that the addition of the 3′- and 5′-terminal regions of the PR8 NA segment might have suppressed the packaging defect to a certain extent.

### 3.3. Growth Kinetics of RG Viruses

MDCK-SIAT1 cells were infected with the rescued RG viruses at a MOI of 0.01 and their growth kinetics were investigated (Figure 4). Sequencing confirmed that the HA and NA vRNA segments did not mutate in the RG viruses. The PR8/Cal, P-Cal-P/PR8, and P-Cal-P/Cal viruses showed growth kinetics and virus titers similar to those of the PR8 virus. In contrast, the PR8/P-Cal virus showed similar growth kinetics for the first 12 h, but reached a plateau of titer ~100-fold lower than that of the PR8 virus. This finding was apparent in the P-Cal-P/P-Cal virus, for which viral growth was significantly delayed and reached a plateau of titer 10,000-fold lower than that of PR8. It is possible that the chimeric NA protein had a kind of functional defect. PR8/115(Cal)100 and Cal-P/115(Cal)100 viral growth was delayed by 12 h, but these viruses reached titers similar to that of the PR8 virus. It is possible, although not proven, that lengthening of the NA vRNA segment caused prolonged genome replication.

### 3.4. Relative Levels of HA and NA in RG Viruses

The RG viral particles were purified from the culture media by ultracentrifugation and analyzed by SDS-PAGE, followed by Western blotting (Figure 5). The blots were probed with anti-H1 HA and anti-NA Abs. When the blots were similarly probed with anti-NP Ab, the levels of the purified viruses, except for the P-Cal-P/P-Cal virus, were found to be largely comparable. Then, the band intensities were semi-quantified using ImageJ software, and the ratios of HA and NA to NP were assessed as the relative levels of HA and NA in the viral particles. The Cal-derived HA level was highest in the P-Cal-P/Cal virus (HA/NP ratio, 0.97), and this was consistent with the result of a study by Harvey et al. [2]. The Cal-derived NA level was highest in the P-Cal-P/Cal mono-chimeric virus (NA/NP ratio, 1.21).

### 3.5. Association of HA and NA with Lipid Rafts

Lipid rafts are defined as membrane fractions of cholesterol- and sphingolipid-enriched membrane microdomains that are insoluble in non-ionic detergent at 4 °C [26]. Previous studies have shown that both HA and NA are associated with lipid rafts, and this association is important for HA/NA membrane transport and subsequent virus release since influenza virus assembles at and buds from lipid rafts [27,28]. To investigate the affinity of Cal HA/NA to the lipid rafts, they were expressed in 293T cells, solubilized with 1% Triton X-100 at 4 °C, and subjected to centrifugation. Triton X-100 solubilization analysis showed that PR8 HA was evenly distributed in detergent-soluble and detergent-insoluble fractions. This distribution pattern was similar to that of authentic Cal and chimeric HAs (Appendix A). When the association of NA to the lipid rafts was investigated, the distribution patterns of PR8, Cal, and P-Cal NAs were found to be not significantly different. These results suggested that the growth efficiency of RG viruses was not attributed to the affinity of their HA/NA to the lipid rafts.

### 3.6. Oligomerization of HA and NA

HA trimers are subjected to conformational changes to allow fold-back of HA intermediates, leading to membrane fusion [29]. Previous studies have shown that the trimer of Cal HA is less stable than those of HAs of other strains, with a narrower pH range for membrane fusion [9,10,11]. We compared the oligomeric states of authentic PR8, Cal, and chimeric HAs by BN-PAGE. Following expression in 293T cells, HA proteins were dissolved by sonication in buffers (pH, 4.5–6.5) and analyzed by BN-PAGE, followed by Western blotting (Figure 6A, left). The PR8 HA trimer was stable at all tested pH values (4.5–6.5). In contrast, the Cal HA trimer was dissociated to a monomer/dimer at pH 6.5. However, the Cal-P HA trimer was largely tolerant to pH 6.5 and only partially dissociated to a monomer/dimer. These results confirmed that the Cal HA trimer was less stable at mildly acidic pH than the PR8 HA trimer, suggesting that chimerization improved the stability of the Cal HA trimer. A single amino acid substitution was observed in the TM domain (see Discussion) of Cal-P HA, which contains the TM-CT regions from PR8. The HA construct of P-Cal-P HA was identical to that of Cal-P HA after cleavage of the signal peptide, since they only differ in the signal peptide region. Since HA trimers form in the endoplasmic reticulum, the trimeric structure of P-Cal-P HA is identical to that of Cal-P HA.

The NA tetramer is composed of two units of intermolecular disulfide-bonded NA dimers. BN-PAGE, followed by Western blotting, revealed that PR8 NA, Cal NA, and P-Cal NA tetramers were stable at pH 4.5–6.5 (Figure 6A, right). When protein samples were analyzed by SDS-PAGE under reducing and non-reducing conditions, PR8, Cal, and P-Cal NAs were resolved to dimers in non-reducing conditions (Figure 6B, right). These results suggested that chimeric NA dimers formed through disulfide bonds, similar to authentic PR8 and Cal NA dimers.

### 3.7. Cell Fusion Ability of HA

Proteolytic cleavage of HA produces HA1 and HA2 subunits, and subsequent exposure to acidic pH induces conformational changes in both subunits, leading to membrane fusion. Cell membrane fusion is critical for multiple rounds of viral replication. Our chimeric HAs, especially P-Cal-P HA, eased the rescue of RG viruses with a PR8 backbone (Figure 2), and the RG viruses produced virus titers comparable with that of the PR8 virus (Figure 4); this was consistent with the result of a previous study [2]. We explored whether this improvement was linked with the cell fusion ability of HA by cotransfecting 293T cells with HA and EGFP expression plasmids and incubating for 24 h. The cells were treated with trypsin and exposed to acidic conditions (pH, 5.0–7.0). Confocal microscopy revealed that PR8 HA-expressing cells induced membrane fusion at pH 5.0–5.6, whereas Cal HA-expressing cells induced membrane fusion at pH 5.0–5.2 (Figure 7), confirming the results of a previous study in which Cal HA induced cell fusion only at a narrow pH range [11]. In contrast, chimeric (Cal-P and P-Cal-P) HA-expressing cells exhibited cell membrane fusion at pH 5.0–5.6, similar to PR8 HA (Figure 7). These results suggested that the chimerization of HA broadened the optimal pH range for HA activation.

### 3.8. NA Activity

The enzymatic activity of NA was investigated by a commonly used assay with 4-MUNANA (Figure 8). The cells transfected with NA expression plasmids were lysed with 0.1% Triton X-100, and the supernatants were subjected to quantitative assay for NA activity. The NA activities were normalized to the amounts of NA protein. PR8 NA was highly active, whereas Cal NA was less active, and this was consistent with the results of a previous study [30]. P-Cal NA was the least active (Figure 8A), as observed in a previous study [4]. We also investigated the NA activities of the RG viruses. The RG virus stocks were similarly lysed with 0.1% Triton X-100 and subjected to NA activity assay. The results were normalized to the NP levels in the virus stocks. Cal NA-containing RG viruses exhibited lower NA activity than PR8 NA-containing RG viruses, and P-Cal NA-containing RG viruses exhibited much lower NA activity (Figure 8B, upper graph). These results suggested that the chimerization of NA impaired NA activity. The infectious titers of the RG virus stocks were measured by plaque assay. Results of the RG viruses were largely consistent with the NA activities, except those with 5′- and 3′-terminal regions added to their NA vRNA segments (Figure 8B, lower graph).

## 4. Discussion

Recent H1N1 vaccine viruses have been produced by coinfection with the Cal-derived virus and a high-yield derivative of the PR8 virus into eggs, followed by negative selection of PR8 HA- and/or PR8 NA-positive viruses. The World Health Organization has recommended the use of PR8 as a backbone virus to generate reassortant viruses for vaccine development [31]. Several studies have suggested that the M segment derived from PR8 is associated with a high-yield phenotype of reassortant viruses [32,33]. A high-throughput mutagenesis study produced high-yield PR8 derivatives, which showed amino acid substitutions in RNA polymerases (PB2, PB1, and PA) [22]. A PR8 derivative that harbors high-fidelity PB1 has also been developed to reduce the amino acid substitutions generated during serial passage in eggs [34]. However, viral growth and HA yield remain concerns for efficient vaccine production. In fact, it has been reported that Cal HA itself is structurally unstable and susceptible to proteolytic digestion [9]. To overcome this insufficiency, several groups have employed site-directed mutagenesis, showing that the amino acid substitutions in HA2 can stabilize the HA trimers and optimize membrane fusion to generate a high-yield virus [10,12,13]. Another study has shown that Cal HA chimeras that contain the N- and/or C-terminal regions of PR8 HA improve viral growth [2]. For the production of these recombinant viruses that harbor genetically engineered HA segments, RG technology must be employed. Systematic analyses of RG viruses will provide a rationale for the construction of a chimeric RG virus vaccine.

In this study, we rescued the RG viruses that harbored chimeric HA/NA constructs using MDCK-SIAT1 cells for virus recovery and evaluated them as cell-based vaccine virus candidates. Our data were consistent with those of a previous study on the egg-grown RG viruses that harbored chimeric HAs [2] (corresponding to our Cal-P and P-Cal-P HAs), confirming that the both-side chimeric HA segment (the 3′-NCR-signal peptide and TM-CT-5′-NCR regions derived from PR8 and the external domain region from Cal) was capable of being incorporated into the RG virus with high virus yield (Figure 2 and Figure 4). The relative HA content in our P-Cal-P/Cal virus was comparable to that in the PR8 virus (Figure 4). Curiously, the Cal/Cal virus (i.e., the RG virus harboring Cal HA and Cal NA on the PR8 backbone) could not be rescued in our 293T/MDCK-SIAT1 cell systems (Figure 2). MDCK cells also failed to produce the Cal/Cal virus. The underlying reasons are currently unknown, but a similar failure in the recovery of this RG virus has been observed in Vero/CHOK1 cell systems [6]. However, successful recovery of this RG virus has been shown in 293T/egg cell systems [2,4].

When the chimeric HA constructs were transfected into MDCK cells and compared with PR8 and Cal HAs, the HA expression levels were found to be comparable (Figure 1). Equivalent HA expression levels were also observed in 293T cells (data not shown). The affinity of these HAs to lipid rafts was comparable (Appendix A). These data suggested that the advantages of chimeric HAs could not be attributed to the HA expression levels or their affinity to lipid rafts. It was more likely that the cell fusion ability of chimeric HAs resulted in these advantages (Figure 7).

HA trimerization was essential for cell membrane fusion in infected cells [7,8]. Our study showed that Cal HA trimer was dissociated at a mildly acidic pH (6.5). In contrast, the chimeric HA, similar to the PR8 HA, was relatively tolerant (Figure 6). These results suggested that the regions replaced by the PR8 HA sequence in the context of the chimeric HAs were responsible for HA trimer stabilization, leading to cell fusion at a broader pH range. It is widely accepted that the signal peptide region is cleaved off before cell membrane targeting of HA, and the membrane fusion ability and trimerization of HA are mostly ascribed to the HA2 subunit. In this study, the TM-CT regions, located within HA2, of Cal HA were replaced with the corresponding regions of PR8. When the amino acid sequences of the TM-CT regions of PR8 and Cal were compared, an amino acid difference (L/V at aa 544) in the TM region was observed (Figure 3), but this substitution did not alter the affinity of HA to lipid rafts (Appendix A). It is possible that a longer side chain of leucine contributed to the stabilization of the TM–TM interactions. As for NA, it has been shown that the TM domain forms a homotetramer and supports NA tetramerization and activity [35].

We also constructed the P-Cal NA, in which the CT-TM-ST domains were derived from PR8 NA and the globular head of NA was from Cal HA. We found that the NA activity of this chimera was significantly reduced (Figure 8). In accordance, P-Cal NA viral growth reached 10^6^ PFU/mL but did not exceed 10^7^ PFU/mL. P-Cal-P HA/P-Cal NA viral growth did not occur for the first 36 h or exceed 10^4^ PFU/mL (Figure 4). From these results, it was very likely that the P-Cal NA construct impaired NA activity. Since the PR8 NA, similarly to the WSN NA, has a large deletion (15 amino acids) in the ST region (compared with other H1N1 viruses, including the Cal virus), the P-Cal NA construct had a considerably shorter ST region than Cal NA. It is possible that the shortening of the ST region led to a failure of multiple rounds of replication of the RG virus, since numerous studies have shown that short ST-containing NA is inefficient in releasing the progeny virus, resulting in a marked reduction in viral growth [36,37]. However, several studies have shown that the RG viruses that harbored P-Cal NA grow well in eggs, to a level similar to that of the parental non-chimeric reassortant virus [4,6]. Although the underlying mechanisms are unknown, cell systems used for virus recovery may have been partly responsible. In H5 and H7 subtypes, viruses isolated from chickens have additional glycosylation sites in HA and a deletion in the ST of NA [38,39]. It has been reported that genetically engineered viruses that contain short ST show better growth efficiency and higher pathogenicity in animals [40,41]. Although the PR8 and Cal HAs have the same number of glycosylation sites, it has been suggested that an optimal balance between receptor binding by HA and receptor cleavage by NA is important for efficient replication of the Cal virus in cells and eggs [30]. Based on this HA/NA balance model, we suggested that the combination of short ST-containing NA with Cal-derived HA exerted negative effects on virus yield in our cell systems. However, other possibilities cannot be ruled out, since amino acid substitutions in the TM-ST regions of PR8 and Cal NAs have been observed.

Each vRNA forms a twisted and helical structure, in which both the 3′- and 5′-terminal ends fold back to form a partially double-stranded panhandle structure that functions as a promoter [18,42]. The sequence of the promoter is highly conserved in the eight vRNAs; however, the flanked nt sequences within the terminal NCRs are less conserved, especially in the HA vRNA (Figure 3). The entire NCR is essential for vRNA packaging and assembly into progeny viruses [18,43,44]. In this study, the one-side chimeric HA (Cal-P HA) very often failed to be rescued, whereas the both-side chimeric HA (P-Cal-P HA) was rescued (Figure 2). It is plausible that the signals for vRNA assembly and packaging were split or not properly formed in the one-side chimeric HA vRNA. In contrast, one-side chimeric NA (P-Cal NA), similar to the authentic Cal NA, was often rescued (Figure 2), possibly because the nucleotide sequences in the 3′- and 5′-terminal NCRs of PR8 and Cal NA vRNAs were highly conserved (Figure 3).

To ensure vRNA packaging, we added the 3′- and 5′-terminal regions of the PR8 HA/NA vRNA segments (100–115 nt) to the HA/NA coding region, respectively (Figure 2). This modification failed to rescue the HA vRNA segment, but this was not the case for the NA vRNA segment. Although there could be a length limitation for the packaging of each vRNA, this is not the case for HA vRNA, since one previous study successfully produced RG viruses that contained foreign gene insertions (corresponding to 90 and 140 amino acids) in the HA vRNA derived from the A/Aichi/2/68 virus [45]. Our RG viruses harboring the NA vRNA added by both the terminal regions (PR8/115(Cal)100 and Cal-P/115(Cal)100 viruses) were recovered, but their growth was slow. In contrast, it is well known that defective interfering particles show rapid growth. Therefore, the lengthening/shortening of vRNA segments might alter the kinetics of RNA replication and viral growth.

## 5. Conclusions

We investigated cell-based vaccine viruses for H1N1pdm09 by RG and provide the following suggestions: (1) in chimeras, the nt sequences of both-side NCRs of the donor virus segment should be matched to those of the backbone virus (PR8); (2) the cell fusion ability at a broad range of pH was conferred by an amino acid substitution in the TM domain; and (3) the addition of the packaging signal regions to the donor HA and NA vRNAs did not improve, but rather delayed, viral growth. These findings can be adopted for designing RG viruses for vaccine development.

## Figures and Tables

**Figure 1 vaccines-08-00458-f001:**
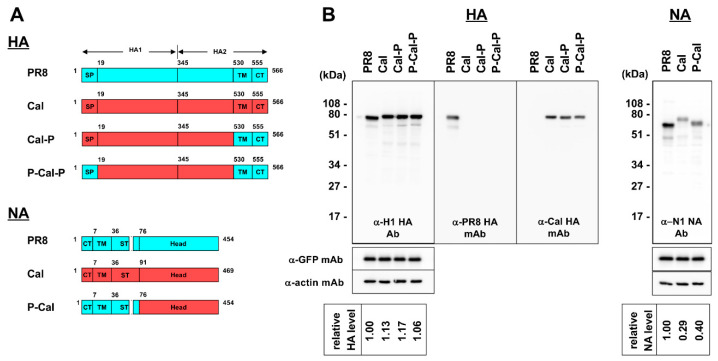
Schematic diagram and expression of chimeric HA and NA proteins. (**A**) Schematic diagram of chimeric HA and NA proteins. Domains derived from A/Puerto Rico/8/34 (H1N1) virus (designated as PR8) are shown as light blue boxes and that of A/California/07/09 (H1N1) virus (designated as Cal) as red boxes. Designations of chimeric HA and NA are shown to the right of constructs. SP—signal peptide; TM—transmembrane; CT—cytoplasmic tail. The amino acid numbers are shown in diagrams. (**B**) Expression of chimeric HA and NA proteins. MDCK cells were cotransfected with an HA or NA expression plasmid and an EGFP expression plasmid. Protein expression was analyzed by Western blotting with anti-H1 HA rabbit Ab, anti-PR8 HA mAb, anti-Cal HA mAb, and anti-NA sheep Ab. The blots were also probed with anti-GFP and anti-actin mAbs. Representative blots are shown. The band intensities were semi-quantified with ImageJ software.

**Figure 2 vaccines-08-00458-f002:**
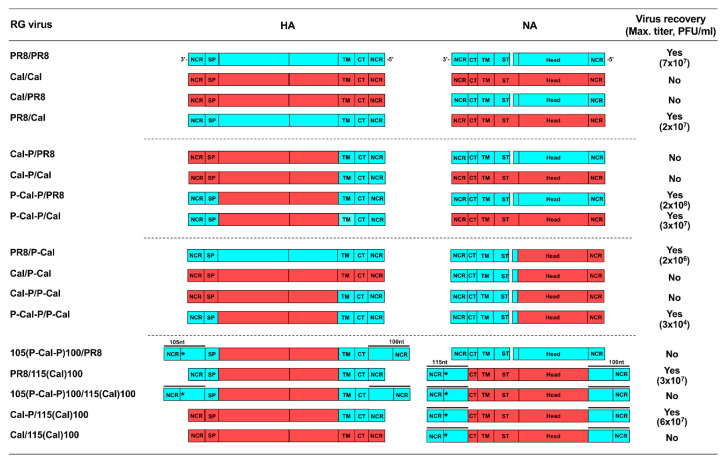
Schematic diagram of RG viruses with chimeric HA and NA segments. Nucleotide sequences derived from PR8 are shown as light blue boxes and sequences from Cal are shown as red boxes. Designations of RG viruses are shown to the left of constructs. For some RG viruses, the 3′- and 5′-terminal sequences (100–115 nt) were added to the coding region (including the start codon and the stop codon) and are shown with overlines. The start codon present in the added regions was substituted with a stop codon (shown with asterisks). NCR—non-coding region; SP—signal peptide; TM—transmembrane; CT—cytoplasmic tail. Recovery of RG viruses was performed in MDCK-SIAT1 cells.

**Figure 3 vaccines-08-00458-f003:**
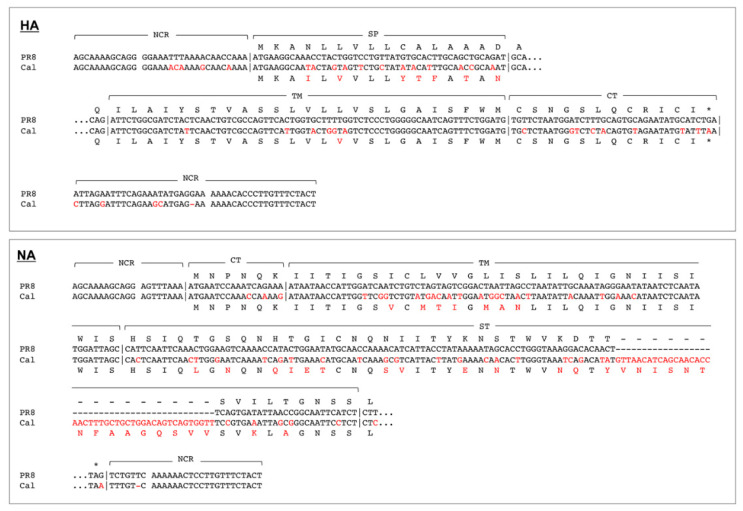
Alignments of nucleotide and amino acid sequences of HA and NA between PR8 and Cal. For HA vRNAs, the 3′-NCR-signal peptide region and the TM-CT-5′-NCR region are shown. For NA vRNAs, the 3′-NCR-CT-TM-ST region and the 5′-NCR region are shown. Nucleotides and amino acids that differ between the two strains are shown in red. Deletion of nucleotides and amino acids is shown with a dash. NCR—non-coding region; SP—signal peptide; TM—transmembrane; CT—cytoplasmic tail.

**Figure 4 vaccines-08-00458-f004:**
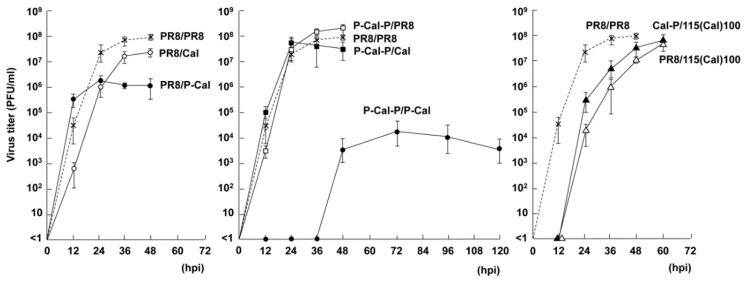
Growth kinetics of RG viruses with chimeric HA and NA. MDCK-SIAT1 cells were infected with RG viruses at a MOI of 0.01 and incubated at 34 °C. The culture media were harvested every 12 h, and virus titers were measured by plaque assay. Data are the means with standard deviations from three independent experiments.

**Figure 5 vaccines-08-00458-f005:**
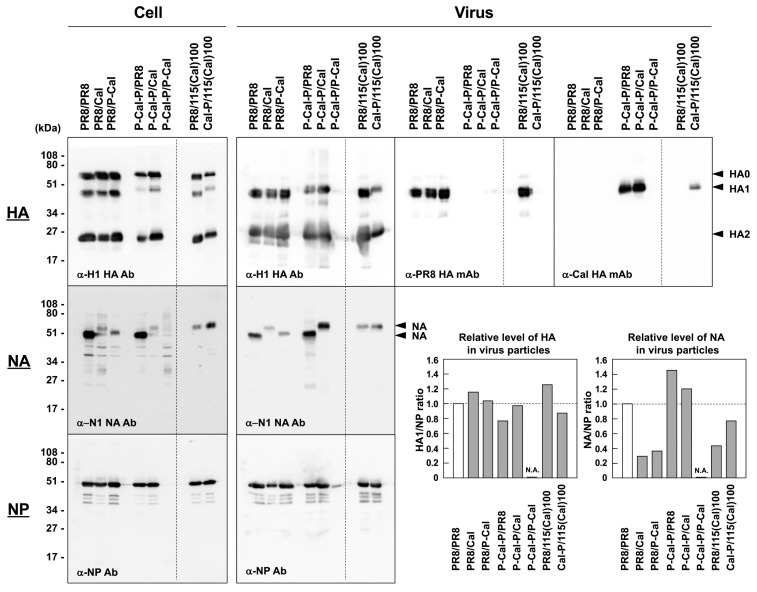
Relative protein levels in RG viruses with chimeric HA and NA. MDCK-1 cells were infected with RG viruses at a MOI of 1.0 and were incubated at 34 °C for 1 d (2 days for P-Cal-P/P-Cal). The RG viruses in the culture media were concentrated through 20% sucrose cushions. The cell and virus samples were subjected to Western blotting with anti-H1 HA rabbit Ab, anti-PR8 HA mAb, anti-Cal HA mAb, anti-NA sheep Ab, and anti-NP rabbit Ab. Representative blots are shown. The band intensities in the virus samples proved by anti-H1 HA rabbit Ab, anti-NA sheep Ab, and anti-NP rabbit Ab were semi-quantified with ImageJ software. The levels of viral particles were normalized to the levels of NP in viral particle fractions. Relative levels of HA/NA in virus particles were evaluated as the ratio of HA/NA to NP. N.A.—not applicable.

**Figure 6 vaccines-08-00458-f006:**
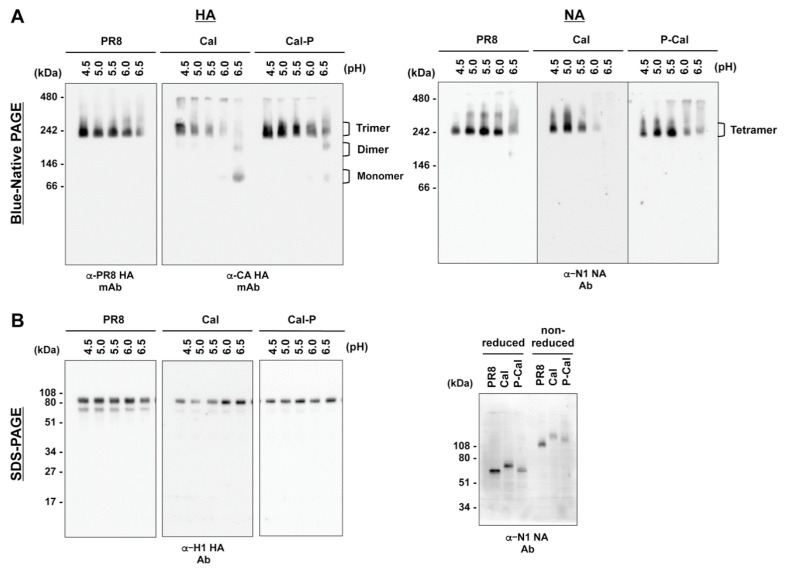
Stability of HA trimers and NA tetramers. 293T cells were transfected with HA and NA expression plasmids. The cells were incubated in buffers (pH, 4.5–6.5) and lysed with 1% Triton X-100. (**A**) After clarification, the supernatants were subjected to Blue-native (BN)-PAGE with 4–12% gradient polyacrylamide gels, followed by Western blotting with anti-PR8 HA or anti-Cal HA mAb (**left**) and anti-NA sheep Ab (**right**). (**B**) The cell lysates were subjected to SDS-PAGE, followed by Western blotting with anti-H1 HA rabbit Ab (**left**) and anti-NA sheep Ab (**right**). Representative blots are shown.

**Figure 7 vaccines-08-00458-f007:**
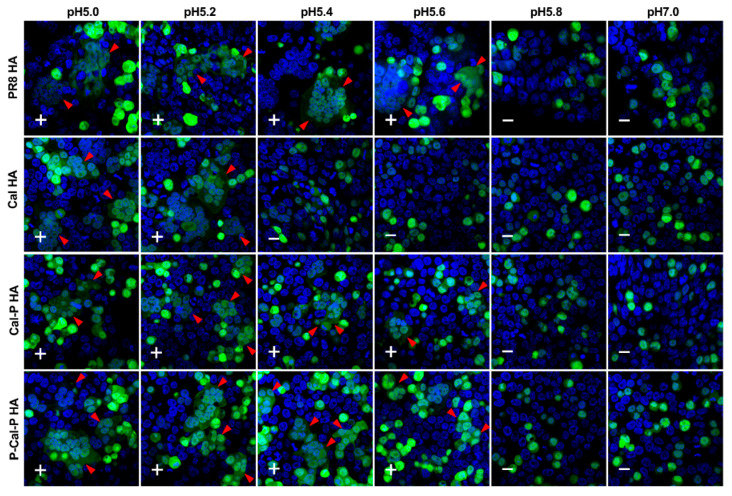
Cell membrane fusion of chimeric HA. 293T cells were cotransfected with HA and EGFP expression plasmids. The cells were treated with acetyl trypsin and subsequently, with buffers of pH 5.0–7.0. The cells were incubated at 37 °C for 2 h. After fixation with paraformaldehyde, cell nuclei were stained with DAPI. Arrowheads indicate cell fusion. +, cell fusion-positive; −, cell fusion-negative.

**Figure 8 vaccines-08-00458-f008:**
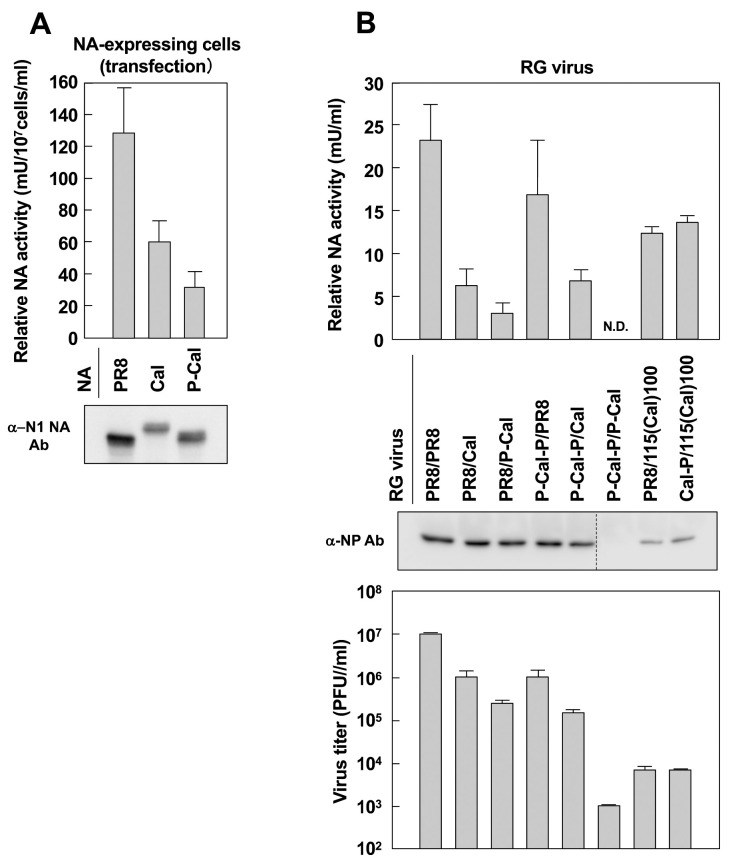
NA activities of chimeric NA and RG viruses and their infectious titers. 293T cells transfected with NA expression plasmids (**A**) and RG virus stocks (**B**) were lysed with 0.1% Triton X-100 and subjected to quantitative NA activity assay with fluorometric 4-MUNANA. Protein expression in cells and RG virus stocks were analyzed by Western blotting with anti-NA sheep Ab (**A**) and anti-NP rabbit Ab (**B**). The NA activity values were normalized to the expression levels of NA (**A**) and the levels of NP in RG viruses (**B**). Data were shown as means with standard deviations from three independent experiments. N.D.—not determined. The infectivity of RG virus stocks was measured by plaque assay in MDCK-SIAT1 cells.

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
