# Peer review of "Cell-Based Influenza A/H1N1pdm09 Vaccine Viruses Containing Chimeric Hemagglutinin with Improved Membrane Fusion Ability"

_vaccines, 2020, doi:10.3390/vaccines8030458_

Round 1

Reviewer 1 Report

Review of manuscript #878146 submitted to Vaccines (MDPI) for publication.

This is a review of the above manuscript submitted by Kawahar et. al for publication in Vaccines. In this manuscript, the authors investigated the replication kinetics and properties of chimeric HA and NA viruses in MDCK-SIAT1 cells. This is critical for ramping up vaccine production especially since vaccine production for pandemic viruses suffers from poor vaccine yield in eggs.

The data suggest that replacing the 3’ and 5’ terminal regions of California pandemic H1N1 with the laboratory strain PR8 improves rescue of the recombinant viruses. The recombinant HAs have better fusogenic activity relative to the wild type HA at a broader pH range. The data shown in this manuscript support previous studies.

Figure 1 Panel A. It will be better to use a clear pattern / higher opacity of color to show different regions of PR8 vs Cal signal peptide, transmembrane region and the cytoplasmic tail.

Line 251. When the authors mention recovery, what HA titer do they mean?

Line 307; were infected to. Please change to third person. MDCK-SIAT1 cells were infected with the rescued RG viruses.

Line 308. Please change to “ Sequencing confirmed that HA and NA vRNA segments did not mutate in the reverse genetics viruses”.

Figure 5.

  1. Please change order of the panels. The NP panel should on top followed by the HA and NA. This is to ensure consistency with the text.
  2. Please indicate HA1 in the HA blots and the band used for comparing NA intensity in the NA blots.
  3. Why were multiple bands observed with NA blots?
  4. Since P-Cal-P RG viruses showed poor NP expression, how were the quantified relative to the other viruses?

Figure 6. Differential insolubility of the HA and NA proteins is not sufficient to conclude that HA and NA in this manuscript are associated with lipid rafts. The authors should show that a known lipid raft associated protein also shows the differential insolubility that they demonstrate for reverse genetics HA/ NA proteins. The authors should consider measuring a GPI anchored protein as a control (since they have been documented to localized to lipid rafts).

Figure 9. Given that NA activity in the reverse genetics viruses is signficantly lower, how do the authors explain the near “wild” type replication of P-Cal-P/ Cal viruses.

Author Response

Comments and Suggestions for Authors

Review of manuscript #878146 submitted to Vaccines (MDPI) for publication.

This is a review of the above manuscript submitted by Kawahara et al for publication in Vaccines. In this manuscript, the authors investigated the replication kinetics and properties of chimeric HA and NA viruses in MDCK-SIAT1 cells. This is critical for ramping up vaccine production especially since vaccine production for pandemic viruses suffers from poor vaccine yield in eggs.

The data suggest that replacing the 3’ and 5’ terminal regions of California pandemic H1N1 with the laboratory strain PR8 improves rescue of the recombinant viruses. The recombinant HAs have better fusogenic activity relative to the wild type HA at a broader pH range. The data shown in this manuscript support previous studies.

Figure 1 Panel A. It will be better to use a clear pattern / higher opacity of color to show different regions of PR8 vs Cal signal peptide, transmembrane region and the cytoplasmic tail.

Re) Following the suggestion, Figures 1A and 2 have now been shown in light blue and red colors.

Line 251. When the authors mention recovery, what HA titer do they mean?

Re) We use “recovery” when infectious virus is rescued. In Figure 2, we added the data of the maximum titers of recovered RG virus in the column of “virus recovery”.

Line 307; were infected to. Please change to third person. MDCK-SIAT1 cells were infected with the rescued RG viruses.

Re) Following the suggestion, we have changed the subject to “MDCK-SIAT cells”.

Line 308. Please change to “ Sequencing confirmed that HA and NA vRNA segments did not mutate in the reverse genetics viruses”.

Re) We appreciated and followed the reviewer’s editing. We asked native English speakers to proofread the entire manuscript.

Figure 5.

1) Please change order of the panels. The NP panel should on top followed by the HA and NA. This is to ensure consistency with the text.

Re 1) We initially followed this suggestion and changed the order of panels. But we found that the figure captions for the two left panels are far away from the panels. Another reviewer also has suggested that the table in Figure 5 is shown in a graph here. Therefore, we changed the text in which HA and NA first appeared and finally NP.

2) Please indicate HA1 in the HA blots and the band used for comparing NA intensity in the NA blots.

Re 2) The positions of HA0, HA1, HA2, and NA have been indicated by arrowheads.

3) Why were multiple bands observed with NA blots?

Re 3) We think that they are most likely breakdown products of NA, especially in cells.

4) Since P-Cal-P RG viruses showed poor NP expression, how were the quantified relative to the other viruses?

Re 4) As the reviewer pointed out, the NP level of the P-Cal-P/P-Cal bi-chimeric virus fraction was very low. Therefore, we have changed the interpretation, suggesting that the quantification of the band intensities were not applicable to this virus. The NP level of P-Cal-P/PR8 and P-Cal-P/Cal viruses was comparable to that of the PR8/PR8 virus.

Figure 6. Differential insolubility of the HA and NA proteins is not sufficient to conclude that HA and NA in this manuscript are associated with lipid rafts. The authors should show that a known lipid raft associated protein also shows the differential insolubility that they demonstrate for reverse genetics HA/ NA proteins. The authors should consider measuring a GPI anchored protein as a control (since they have been documented to localized to lipid rafts).

Re) We accept the reviewer’s comments and have changed Figure 6 to supplementary data, since the data shown are not equilibrium flotation centrifugation in gradients. But we included the data on caveolin (as raft-associated protein) and p75 (as non-raft marker) in Figure 1S. We routinely detect caveolin and p75 or transferrin receptor in detergent-insolubility experiments.

Figure 9. Given that NA activity in the reverse genetics viruses is significantly lower, how do the authors explain the near “wild” type replication of P-Cal-P/Cal viruses.

Re) HA plays a role in virus entry to host cells, whereas NA plays a role in release from the cells (but not in virus entry). Although the Cal-derived NA showed relatively low NA activity, this does not matter at the stage of virus entry and not much in a single round of virus replication. The defective phenotype due to the low NA activity will become apparent in a multiple round of viral replication. If cells were infected with the P-Cal-P/Cal virus at a much lower MOI, we would be able to see a delay in virus growth. Another possibility would be that the inactivation kinetics of NA activity may differ in the NA constructs. We observe that the NA activity is not so stable. Alternatively, the enzymatic activity of P-Cal-P/Cal NA may have been sufficient for and above the threshold for exhibiting the function of NA.

Reviewer 2 Report

The paper entitled “Findings for Development of Cell-based Vaccine Viruses of Influenza A/H1N1pdm09 Containing Chimeric Hemagglutinin and Neuraminidase” authored by  Kawahara, M et al describes interesting observations regarding the rescue of recombinant influenza viruses displaying hybrid glycoproteins using a vaccine platform. Indeed, the need for a Flu vaccine based on reverse genetics and cell culture production has a high priority for epidemics preparedness. Notwithstanding, the paper  in its present form has important methodological and style issues that precludes me from recommending publication in Vaccines.

This paper described interesting in vitro observations to characterize potential Flu vaccine candidates, but there is no immunogenicity experiment.

 Major points

  1. The tittle needs to be declarative. There is no hint for results or conclusions and as a results the paper reads as a mix of a review where the authors original work is hard to discern immediately. For instance in line 56 and 57 it reads: “To further improve the virus growth and/or HA yield, the recombinant viruses containing chimeric HA between Cal and PR8 were generated by RG” and what follows is a description of reference publications. Indeed the authors did RG on influenza viruses but it is not easy to figure out what exactly they did. Scientific English style review will greatly benefit the manuscript.
  2. The generation of MDCK-SIAT1 cells was reproduced by the authors, however there is no indication of the method, the plasmid map, its sequence deposited in Genebank, how the cells were maintained, if they depend on continuous hygromycin selection, how they were used to amplify the RG generated flu vaccines, the stable expression of SIAT1, among other characteristics that are paramount to support their use for vaccine production. If not for vaccine production, then an approved cell line for vaccine production has to be used to explore multi-step growth kinetics.
  1. Construction and expression of chimeric HA and NA was conducted using semiquantitative western blots, however the proper loading controls are missing. How to be sure that the observed differences are not due to transfection efficiency?
  2. Oligomerization of HA and NA. This analysis must be performed using purified, concentrated and quantified RG generated viral particles (or infectious units), otherwise the described observations can be due to transfection efficiency. Why did the authors use purified viral particles to determine the rate of glycoproteins incorporation and not here?
  3. There is no relevant information coming from the association of HA and NA with lipid rafts as shown. The technique used by the authors is an abbreviated method, there is no raft protein marker to ascertain the fractionation was successful and this biochemical approach have a lot of detractors that deem it just a biochemical artifact. I suggest to drop these results or to complete this analysis comprehensibly (lipid raft markers and a proper density fractionation gradient).
  4. The proportion of infectious to total particles for the most significant recombinant viruses need to be analyzed.

Minor points

  1. Rescue of RG viruses containing chimeric HA/NA. the table depicted in figure 2 needs to be completed using a quantitative parameter of virus replication, such as maximum titer observed. The growth kinetic is a bit far removed from this table.
  2. The table included in figure 5 would be read easier with a bar graph instead. It would be a better representation of the observed results. The amounts of infectious units (or viral particles if available) needs to be specified in materials and methods and the figure legend.
  3. What about the cell membrane fusion of PR8/115(Cal)100 and Cal-P/115(Cal)100 viruses?
  4. Grammatical mistakes and suboptimal sentence construction is noted. The paper needs to be proof read thoroughly.

Author Response

Comments and Suggestions for Authors

The paper entitled “Findings for Development of Cell-based Vaccine Viruses of Influenza A/H1N1pdm09 Containing Chimeric Hemagglutinin and Neuraminidase” authored by Kawahara, M et al describes interesting observations regarding the rescue of recombinant influenza viruses displaying hybrid glycoproteins using a vaccine platform. Indeed, the need for a Flu vaccine based on reverse genetics and cell culture production has a high priority for epidemics preparedness. Notwithstanding, the paper in its present form has important methodological and style issues that precludes me from recommending publication in Vaccines.

This paper described interesting in vitro observations to characterize potential Flu vaccine candidates, but there is no immunogenicity experiment.

Re) We apologize the omission of information on establishment of MDCK-SIAT1 cells. We have added the details to the M&M section. We rewrote the text and asked native English speakers to proofread the entire manuscript.

Major points

1) The title needs to be declarative. There is no hint for results or conclusions and as a results the paper reads as a mix of a review where the authors original work is hard to discern immediately. For instance in line 56 and 57 it reads: “To further improve the virus growth and/or HA yield, the recombinant viruses containing chimeric HA between Cal and PR8 were generated by RG” and what follows is a description of reference publications. Indeed the authors did RG on influenza viruses but it is not easy to figure out what exactly they did. Scientific English style review will greatly benefit the manuscript.

Re 1) We admit the reviewer’s arguments and have changed the title to “Cell-based Influenza A/H1N1pdm09 Vaccine Viruses Containing Chimeric Hemagglutinin with improved Membrane Fusion Ability”. We deleted the irrelevant information from the introduction and instead stated clearly what we did in this study.

2) The generation of MDCK-SIAT1 cells was reproduced by the authors, however there is no indication of the method, the plasmid map, its sequence deposited in Genebank, how the cells were maintained, if they depend on continuous hygromycin selection, how they were used to amplify the RG generated flu vaccines, the stable expression of SIAT1, among other characteristics that are paramount to support their use for vaccine production. If not for vaccine production, then an approved cell line for vaccine production has to be used to explore multi-step growth kinetics.

Re 2) We apologize the omission of information on our SIAT1 expression plasmid and establishment of MDCK-SIAT1 cells. The details have been added to the M&M (lines 107-123 and 143-148). Our MDCK-SIAT1 cells are maintained in the absence of hygromycin B after cell establishment.

4) Construction and expression of chimeric HA and NA was conducted using semiquantitative western blots, however the proper loading controls are missing. How to be sure that the observed differences are not due to transfection efficiency?

Re 4) In transfection experiments, we routinely use equivalent amounts of DNA (below the saturation level) when expression levels of protein are compared. We also often use two plasmids, a plasmid of interest plus a control plasmid (e.g., GFP or b-Gal expression plasmid) to confirm equivalent transfection efficiencies. Upon the reviewer’s request, we replaced the western blots in Figure 1 by the blots of co-transfection with an HA/NA expression plasmid and an EGFP expression plasmid. We have also added the blots for actin (as loading controls).

5) Oligomerization of HA and NA. This analysis must be performed using purified, concentrated and quantified RG generated viral particles (or infectious units), otherwise the described observations can be due to transfection efficiency. Why did the authors use purified viral particles to determine the rate of glycoproteins incorporation and not here?

Re 5) We thank you for the suggestion. We initially thought this suggestion reasonable but later found it would be difficult to carry out, especially for HA, because of the reasons below. It would be greatly appreciated if we could have your suggestions.

  1. The RG viruses we rescued did not include the Cal/Cal virus (The Cal/Cal virus was not rescued). HA trimerization can be compared between the PR8/PR8 and P-Cal-P/PR8 viruses (both are expected to have relatively stable HA trimers), but the data of Cal/Cal virus (unstable trimer) is lacking.
  2. The virus particles used in Figure 5 were purified from infected cells in presence of trypsin and thus infectious. As you can see, the HA in the virus particle fractions is mostly cleaved to HA1 and HA2. If we used these samples for native-PAGE, we will see lots of bands: not only the monomer, dimer, and trimer of HA1 but also those of HA2. If HA1 and HA2 are associated in a complex form, more bands appear. It is probable that HA1 and HA2 are partially dissociated from the complex form. This condition would not allow us to identify which of these bands is what. Alternatively, it may be possible to use the virus particles purified in absence of trypsin (i.e., uninfectious particles) although the virus yield was perhaps low.
  3. We searched the references for biochemical analysis of HA trimerization and NA tetramerization. One study used 1 ug of recombinant HA purified from insect cells (Feshchenko et al, BMC Biotechnol, 12:77, 2012), and another used radioisotope-labeled cells in absence of trypsin (Magadan et al, J Virol, 87:9742-9753, 2013). Transfected cells were also used for analysis of NA tertramerization (da Silva et al, JBC 288:644-653, 2013). As far as our knowledge, purified virus particles have not been subjected to this assay, perhaps because of the detection limit.

6) There is no relevant information coming from the association of HA and NA with lipid rafts as shown. The technique used by the authors is an abbreviated method, there is no raft protein marker to ascertain the fractionation was successful and this biochemical approach have a lot of detractors that deem it just a biochemical artifact. I suggest to drop these results or to complete this analysis comprehensibly (lipid raft markers and a proper density fractionation gradient).

Re 6) We accept the reviewer’s comments and have changed Figure 6 to supplementary data, since the data shown are not equilibrium flotation centrifugation in gradients. But we included the data on caveolin (as raft-associated protein) and p75 (as non-raft marker) in Figure 1S. We routinely detect caveolin and p75 or transferrin receptor in detergent-insolubility experiments. We also added some text explaining why we looked at the association with lipid rafts.

7) The proportion of infectious to total particles for the most significant recombinant viruses need to be analyzed.

Re 7) We have added the data of the infectious titers of RG virus stocks, which were used for western blotting for NP in Figure 8.

Minor points

1) Rescue of RG viruses containing chimeric HA/NA. The table depicted in figure 2 needs to be completed using a quantitative parameter of virus replication, such as maximum titer observed. The growth kinetic is a bit far removed from this table.

Re 1) The maximum titers of the rescued RG viruses were added to Figure 2.

2) The table included in figure 5 would be read easier with a bar graph instead. It would be a better representation of the observed results. The amounts of infectious units (or viral particles if available) needs to be specified in materials and methods and the figure legend.

Re 2) We appreciate the reviewer’s suggestion, and the HA/NP and NA/NP ratios are now shown in graphs. We added the methods for normalization of the amounts of viral particles to the M&M section and the figure legend.

3) What about the cell membrane fusion of PR8/115(Cal)100 and Cal-P/115(Cal)100 viruses?

Re 3) The PR8/115(Cal)100 virus showed a membrane fusion phenotype very similar to PR8. We did not test the Cal-P/115(Cal)100 virus.

4) Grammatical mistakes and suboptimal sentence construction is noted. The paper needs to be proof read thoroughly.

Re 4) Following the suggestion, we requested native speakers of English to proofread our English writing.

Round 2

Reviewer 2 Report

This revised version, now entitled "Cell-based Influenza A/H1N1pdm09 Containing Chimeric HemagglutininMembrane Fusion Ability" presents in a more robust, comprehensive and readable format the research that  authors have executed.

In my opinion, there is still a possibility to conduct the HA, NA interaction study using viral particles. I understand the hurdles of the setting the authors mention. However if each of the surface proteins are tagged differentially, then a selective WB can recognize easily multimers on a native gel. However, I am also convinced that maybe this analysis may be part of  another study.

I commend the authors for they promptness in their response to comments.